# Mating Competitiveness of Male *Spodoptera frugiperda* (Smith) Irradiated by X-rays

**DOI:** 10.3390/insects14020137

**Published:** 2023-01-29

**Authors:** Shan Jiang, Xiao-Ting Sun, Shi-Shuai Ge, Xian-Ming Yang, Kong-Ming Wu

**Affiliations:** 1College of Plant Protection, Shenyang Agricultural University, Shenyang 110866, China; 2State Key Laboratory for Biology of Plant Diseases and Insect Pests, Institute of Plant Protection, Chinese Academy of Agricultural Sciences, Beijing 100193, China; 3College of Tropical Crops, Hainan University, Haikou 570228, China; 4State Key Laboratory of Ecological Pest Control for Fujian and Taiwan Crops, Institute of Applied Ecology, Fujian Agriculture and Forestry University, Fuzhou 350002, China

**Keywords:** sterile insect technique, *Spodoptera frugiperda*, release ratio, mating competitiveness, control effect

## Abstract

**Simple Summary:**

The invasive pest *Spodoptera frugiperda* has become a global problem. Sterile insect technology is an effective means to prevent invasive insects, but at present, the research on this pest is not perfect. Our results showed that egg sterility in offspring reached 74% when the ratio of those given a 250 Gy dose of X-ray radiation to the non-irradiated males was 12:1, and there was no significant difference in mating competitiveness between irradiated males of different ages. Additional cornfield cage studies confirmed this finding by showing that release ratios from 12:1 to 20:1 significantly reduced larval populations in the field, with a 48 to 69% leaf protection effect and a 58 to 83% insect population reduction. This study provides fundamental information for the management of *S. frugiperda* through the use of sterile insect techniques.

**Abstract:**

*Spodoptera frugiperda*, an invasive pest, has a huge impact on food production in Asia and Africa. The potential and advantages of sterile insect techniques for the permanent control of *S. frugiperda* have been demonstrated, but the methods for their field application are still unavailable. For the purposes of this study, male pupae of *S. frugiperda* were irradiated with an X-ray dose of 250 Gy to examine the effects of both the release ratio and the age of the irradiated males on the sterility of their offspring. The control effect of the irradiated male release ratio on *S. frugiperda* was evaluated using field-cage experiments in a cornfield. The results showed that when the ratio of irradiated males to non-irradiated males reached 12:1, the egg-hatching rate of the offspring of *S. frugiperda* decreased to less than 26%, and there was also no significant difference in mating competitiveness among the different ages. Field-cage testing showed that when irradiated males were released at ratios of 12:1–20:1 to normal males, the leaf protection effect for the corn reached 48–69% and the reduction in the insect population reached 58–83%. In this study, an appropriate release ratio is suggested, and the mating competitiveness of irradiated and non-irradiated males of *S. frugiperda* is investigated, thus providing a theoretical basis for the use of sterile insect techniques to control *S. frugiperda*.

## 1. Introduction

The fall armyworm, *Spodoptera frugiperda* (Smith) (Lepidoptera: Noctuidae), is a tropical and subtropical pest native to the Americas that can damage corn, wheat, and other major crops [1,2], is well known for its polyphagia, high fecundity, and strong migration ability. During the reproductive period, female *S. frugiperda* can mate with different males [1]. A female of *S. frugiperda* can lay about 1000 eggs in 4 to 9 days, and at the proper temperature (20–30 °C), these eggs hatch in 2 to 4 days. The damage to the host plant is made worse as the larvae reach their sixth instar because their greedy appetite is completely displayed [3,4]. Following its invasion of West Africa in 2016, *S. frugiperda* quickly spread to numerous countries in Africa and Asia, where it now poses a serious threat to global food security [5]. Yields can be reduced by more than 70% when corn is severely damaged, with an average reduction of 20% to 40% [6,7,8]. Chemical insecticides were the main method for emergency control after *S. frugiperda* invaded Africa and Asia. Three times as much insecticide is now used compared to before the invasion of *S. frugiperda* [9]. The increasing use of chemical pesticides has undoubtedly improved the resistance of *S. frugiperda*. As *S. frugiperda* began to develop resistance to organophosphates, carbamate, pyrethroid, benzoylurea, and other insecticides in the 1980s, numerous production and ecological issues emerged [10,11,12,13]. Zhao et al. (2019) examined the effects of 21 common chemical insecticides on the *S. frugiperda* population in China. The results showed that traditional pesticides such as carbamate, pyrethroid, and organophosphorus could not control *S. frugiperda* effectively [14]. Meanwhile, the resistance levels of the invasive Chinese *S. frugiperda* were evaluated by Zhang et al. (2020), who found that this pest has developed medium-to-high levels of resistance to pyrethroids, organophosphates, and spinosad [15]. Therefore, the research and development of new, green control technologies are essential for the prevention and control of *S. frugiperda* [9].

An efficient method to control invasive pests is the sterile insect technique (SIT). The purpose of this method is to control the target pest by mating sterile males with wild females. The theory behind it is that the chromosomes of insects are damaged by gamma rays, X-rays, or high-energy electron beam irradiation, causing reproductive damage and sterility [16]. The first large-scale usage of sterile insect technology occurred in the 1950s and was successful in removing the negative effects of *Cochliomyia hominivorax* (Coquerel) in the southeastern United States [17,18]. Since then, a number of invasive pests have been successfully eliminated using this technology, including *Ceratitis capitata* (Wiedemann), which was first discovered in Mexico in 1982 [19]. In 1993, Japan declared that *Bactrocera cucurbitae* (Coquillett) had been successfully eradicated [20,21]. Scientists also employed insect sterility technology in 1989 and 1990 to eradicate Australian *Bactrocera tryoni* (Froggatt) [22].

Due to the radiation tolerance of members of Lepidoptera, the development of sterile insect technology for their control was later than that of Diptera [23]. Higher radiation doses are typically required to sterilize these types of insects, which can significantly reduce their sexual competitiveness and thus negatively impact the success of an insect sterility program. When low-dose radiation-treated male insects mate with non-irradiated females, most of the offspring die, and the survivors also inherit sterile genes. This technique is known as inherited sterility (IS) [24]. Inherited sterility has been used to control various lepidopteran pests. For example, the sterile insect technique was used to successfully manage *Cydia pomonella* (L.) in orchards in the central and southern regions of British Columbia in Canada [25,26]. Before the release of irradiated male insects to control *Pectinophora gossypiella* (Saunders), *Bacillus thuringiensis* (Bt) cotton was used to decrease the density of the insect population in North America. These pests have been successfully eliminated using this method throughout the continental United States and in northern Mexico [27,28].

When using sterile insect techniques, the size of the release area depends on the migratory distribution of the wild population. Projects will be more difficult with long-distance migrating insects because they often require wider release areas [29]. Due to the fact that *S. frugiperda* cannot complete its annual life cycle at temperatures below 15 °C, only the tropical and subtropical regions of China are suitable for its annual breeding cycle (such as Hainan, Guangxi, Yunnan, etc.) [4]. As a result, it does not survive the winter in most of China, the Korean Peninsula, or the temperate regions of Japan [30]. *Spodoptera frugiperda* mostly migrates with the monsoon in the spring and fall, and its winter distribution may be limited due to its inability to migrate and spread during the winter, which makes sterile insect techniques appropriate [30]. Currently, China has approved the commercial planting of Bt maize [31]. If Bt maize is planted in the annual breeding area of the species during the winter, and sterile insects are released there, it is possible to control the density of *S. frugiperda* in the source area and significantly reduce the number of adults that migrate north in spring each year [32].

The sexual competitiveness of irradiated insects is essential for sterile insect technology. Irradiated males must typically be tested for mating competitiveness prior to their widespread release [33,34]. Research has shown that irradiated males are less competitive than normal males. Studies have shown that males exposed to high doses of irradiation in some species are less competitive than normal males, however, this competitive difference can be compensated for by increasing the release ratio [35,36,37]. Fried (1971) established a mathematical model to predict the relationship between the release ratio and competitiveness, and he showed that, in theory, increasing the release ratio could successfully control the pest population [38]. According to previous studies on mosquitoes, infertile *Anopheles arabiensis* (Patton) males exhibited good mating competitiveness when they were three times greater in number than normal males [39]; likewise, it has been reported that *Aedes albopictus* (Skuse) and *Aedes aegypti* (L.) have good control effects when the number of sterile males is 10 times that of normal males [40]. 

In our previous study [32], the 8-day-old male pupae of *S. frugiperda* were exposed to X-rays at dosages ranging from 50 to 400 Gy, and 250 Gy dose had no significant effect on biological parameters such as emergence and life span of male pupa, but it could make the egg sterility rate reach 85%. In this study, male pupae of *S. frugiperda* were irradiated with a dose of 250 Gy to assess their ability to compete for mating in a laboratory environment. The field control effects of releasing sterile males were then investigated in order to determine the ideal release ratio to promote the development and application of sterile insect techniques.

## 2. Materials and Methods

### 2.1. Methods of Irradiating and Feeding Tested Insects

The larvae of *S. frugiperda* were collected from a corn field in Dehong Dai and Jingpo Autonomous Prefecture, Yunnan Province, in January 2019. The first instar larvae were placed in plastic boxes (22 cm × 15 cm × 8 cm) and raised on an artificial diet consisting of soybean flour and wheat flour [41]. There were 30 to 50 larvae per box, and they were transferred to plastic trays containing vermiculite when they were about to pupate. The male and female pupae were separated on the fifth day after pupation (the female pupa has a median longitudinal slit on the eighth abdominal sternite, while both sides are flat, without any protuberances; the male pupa has the eighth abdominal sternite unsplit, and the ninth sternite with a median longitudinal slit and a semicircular protuberance on either side) [42] and placed in separate 30 cm × 30 cm × 30 cm sarongs for emergence. After the emergence of the adult insects, the male and female insects were paired in a bucket (diameter of 25 cm and height of 30 cm, 15 L), the mouth of the bucket was closed with gauze, and the insects were fed with cotton balls dipped in 5% *v*/*v* honey water every day to supplement their nutrition. The eggs were collected after the female insects laid them and were placed in airtight plastic bags for hatching. The feeding conditions for all insect states were as follows: temperature 25 °C ± 1 °C, relative humidity 70% ± 5%, photoperiod L: D = 16 h: 8 h, and the feeding environment was an artificial climate incubator (MGC-450HP, Shanghai Yiheng Scientific Instrument Co., Ltd., Shanghai, China).

The irradiation source used was an X-ray irradiator (X-ray RAD 320, Precision X-ray Inc., Branford, CT, USA). The radiation dose rate was 4.67 Gy/min (320 kV, 12.5 mA, 40 cm SSD, HVL = 1 mm Cu). The insects were illuminated on a lead table in the center of the irradiator. The irradiation was stopped and samples were collected when the total cumulative radiation dosage reached 250 Gy (3212 s). The used insects were male pupae that were 8 days old (1–2 days before emergence) [24,32,43] and exposed to radiation at ambient temperature. 

### 2.2. Field Corn Cage Coverage Methods

The experimental field was located at the Xinxiang Comprehensive Experimental Base of the Chinese Academy of Agricultural Sciences in Qiliying Town, Xinxiang County, Xinxiang City, Henan Province (35°8′49.308″ N; 113°47′21.12″ E). The experimental field was divided into 24 plots, and the cage size of each plot was 6 m × 6 m × 4 m. The total area of the experimental field was 0.618 hm^2^. The cages of the different treatment plots were separated by 2.5–5 m and were randomly arranged. After the cages were built in the field, maize (Longxiang 111, Hebei Keteng Biotechnology Co., Ltd., Shijiazhuang, Hebei, China) was planted in the cages, with a plant spacing of 54 cm and a row spacing of 50 cm, and 90 corn plants were planted in each cage. The experiment started on 7 September 2022, when the corn was at the V6 growth stage (sixth vegetative leaf), and ended on 30 September 2022.

### 2.3. Experimental Methods

#### 2.3.1. Effect of the Release Ratio of Irradiated to Non-Irradiated Males on the Sterility Rate of the Offspring

The male pupae of *S. frugiperda* were irradiated with 250 Gy of X-rays. After emergence, the irradiated and non-irradiated insects were placed into a rearing bucket (diameter of 25 cm and height of 30 cm, 15 L) in the following proportions of irradiated males: non-irradiated males: non-irradiated females = 0:1:1 (control), 2:1:1, 4:1:1, 6:1:1, 8:1:1, 10:1:1, 12:1:1, 16:1:1, and 20:1:1 (there was one non-irradiated female and one non-irradiated male insect in each treatment, the irradiated males were released proportionally on this basis, and all insects were one day old) [44,45]. The feeding methods and the environment in which the insects were kept were described in Section 2.1. Each treatment was repeated 20 times. Eggs were collected daily, and the number of eggs laid and the hatching rate was recorded. The competition of irradiated males was determined by calculating the sterility rate (the probability that the eggs will not hatch after the females lay the eggs).

#### 2.3.2. Selection of the Age of Irradiated Males at Release

Male pupae were irradiated with 250 Gy. On days 1 (control), 2, 3, and 4 after emergence, the irradiated males were placed in a bucket containing one non-irradiated female and one non-irradiated male (these non-irradiated insects were both one day old). The mating ratio was as follows: irradiated males: non-irradiated males: non-irradiated females = 12:1:1. The insects were kept in buckets (diameter of 25 cm and height of 30 cm, 15 L) after pairing, egg-laying, and hatching and were observed daily, and the infertility rate of the eggs was used to assess the effectiveness of control. The feeding methods and the environment in which the insects were kept were described in Section 2.1. Each treatment was repeated 15 times.

#### 2.3.3. Confined-Field Release Test of Irradiated Males

Male pupae of *S. frugiperda* were irradiated with 250 Gy of X-rays and released proportionally to non-irradiated insects in a field cage made with 40 mesh white polyethylene after the irradiated males emerged (irradiated males: non-irradiated males: non-irradiated females = 0:1:1 (control), 1:0:1, 12:1:1, 16:1:1, 20:1:1, 0:1:1 (spraying insecticide)). The insecticide used in the experiment was 8% emamectin benzoate (Hebei Bojia Agricultural Co., Ltd., Shijiazhuang, Hebei, China), which was sprayed evenly using an electric sprayer on 10 September 2022 at a concentration of 0.15 g/L. Ten non-irradiated females and ten non-irradiated males were eventually released in each cage, and the irradiated males were released proportionally on this basis (the number of insects: 0:1:1 (control) = 0:10:10, 1:0:1 = 10:0:10, 12:1:1 = 120:10:10, 16:1:1 = 160:10:10, 20:1:1 = 200:10:10, 0:1:1 (spraying insecticide) = 0:10:10). The experiment was repeated four times.

On days 3, 5, 7, 9, 14, and 21 after the release of the insects, all plants in the cage were surveyed; the leaves containing eggs were marked, and the number of eggs was recorded. The number of unhatched eggs on marked leaves was recorded in follow-up surveys, and the non-hatching rate was calculated; the number of damaged plants was also recorded, and the leaf damage index was calculated. Three weeks after insect release (21 days), the insect population in the field was surveyed, and the corrected rate of insect decline and the corrected rate of leaf protection was calculated.
(1)$$\mathrm{Damaged}\;\mathrm{leaf}\;\mathrm{index}=\frac{\sum (\mathrm{number}\;\mathrm{of}\;\mathrm{damaged}\;\mathrm{leaves}\;\mathrm{at}\;\mathrm{each}\;\mathrm{level}\;\times \;\mathrm{each}\;\mathrm{level})}{\left(\mathrm{total}\;\mathrm{number}\;\mathrm{of}\;\mathrm{investigated}\;\mathrm{leaves}\right)\times \;\mathrm{highest}\;\mathrm{level}}\times 100\%$$
(2)$$\mathrm{Corrected}\;\mathrm{non-hatching}\mathrm{hatching}\;\mathrm{rate}\;\mathrm{of}\;\mathrm{eggs}=\frac{\mathrm{Infertility}\;\mathrm{rate}\;\mathrm{in}\;\mathrm{the}\;\mathrm{treatment}\;\mathrm{plot}\;-\;\mathrm{Infertility}\;\mathrm{rate}\;\mathrm{in}\;\mathrm{the}\;\mathrm{control}\;\mathrm{plot}}{1-\mathrm{Infertility}\;\mathrm{rate}\;\mathrm{in}\;\mathrm{the}\;\mathrm{control}\;\mathrm{plot}}\times 100\%$$
(3)$$\mathrm{Corrected}\;\mathrm{leaf}\;\mathrm{protection}\;\mathrm{rate}=\frac{\mathrm{damaged}\;\mathrm{leaf}\;\mathrm{index}\;\mathrm{in}\;\mathrm{the}\;\mathrm{control}\;\mathrm{plot}-\mathrm{Damaged}\;\mathrm{leaf}\;\mathrm{index}\;\mathrm{in}\;\mathrm{the}\;\mathrm{treatment}\;\mathrm{plot}}{\mathrm{Damaged}\;\mathrm{leaf}\;\mathrm{index}\;\mathrm{in}\;\mathrm{the}\;\mathrm{control}\;\mathrm{plot}}\times 100\%\;$$
(4)$$\mathrm{Population}\;\mathrm{decline}\;\mathrm{rate}=\frac{\mathrm{number}\;\mathrm{of}\;\mathrm{insects}\;\mathrm{in}\;\mathrm{the}\;\mathrm{control}\;\mathrm{plot}\;-\mathrm{number}\;\mathrm{of}\;\mathrm{insects}\;\mathrm{in}\;\mathrm{the}\;\mathrm{treatment}\;\mathrm{plot}}{\mathrm{number}\;\mathrm{of}\;\mathrm{insects}\;\mathrm{in}\;\mathrm{the}\;\mathrm{control}\;\mathrm{plot}}\times 100\%\;$$
(5)$$\mathrm{Correction}\;\mathrm{of}\;\mathrm{population}\;\mathrm{decline}\;\mathrm{rate}=\frac{\mathrm{population}\;\mathrm{decline}\;\mathrm{rate}\;\mathrm{of}\;\mathrm{the}\;\mathrm{treatment}\;\mathrm{plot}-\mathrm{population}\;\mathrm{decline}\;\mathrm{rate}\;\mathrm{of}\;\mathrm{the}\;\mathrm{control}\;\mathrm{plot}}{1-\mathrm{population}\;\mathrm{decline}\;\mathrm{rate}\;\mathrm{of}\;\mathrm{the}\;\mathrm{control}\;\mathrm{plot}}\times 100\%$$

The leaf damage index was divided into six levels based on the percentage of the total leaf area that was damaged: 0%, 1–20%, 21–40%, 41–60%, 61–80%, and 81–100% [46].

### 2.4. Data Analysis

The number of eggs laid, infertility rate, hatching rate, leaf damage index, and population decline rate of *S. frugiperda* under different release ratios were analyzed by one-way ANOVA, and Tukey’s HSD was used for multiple comparisons if the difference was significant. One-way analysis of variance was used to analyze the effect of the age of irradiated males on the egg infertility rate of *S. frugiperda*, and Tukey’s HSD multiple comparisons were performed if the difference was significant. Percentage data were arcsine square root transformed before ANOVA. All data analyses were performed in SPSS 23.0 software (SPSS Software Inc., Chicago, IL 60076, USA). Figures were plotted in GraphPad 8.0 (GraphPad Software Inc., San Diego 92121, CA). With the use of Origin (2019) software (Origin Lab Corporation, Northampton, MA 02115, USA), the logistic curve was used to fit and plot the infertility rate of *S. frugiperda* at various release ratios. The model equation was as follows: $y=\frac{k}{1+e^{a-rx}}$, where *y* represents the infertility rate, *k* represents the highest theoretical percentage, *x* represents the radiation dose, *r* represents the growth rate coefficient, and *a* is the shape parameter.

## 3. Results

### 3.1. Effect of the Release Ratio of Irradiated to Non-Irradiated Males on the Sterility Rate of Offspring

One-way analysis of variance showed that different release ratios had no significant effect on the number of eggs laid (*F*_8,171_ = 0.647, *p* = 0.737, Figure 1A), and different release ratios had a significant effect on the infertility rate of the insects (*F*_8,171_ = 9.471, *p* < 0.001, Figure 1B). The infertility rates for the 10:1, 12:1, 16:1, and 20:1 groups (irradiated: non-irradiated males) (70.59–81.57%) were significantly higher than that of the control (17.67%) (Figure 1B). The logistic function was the equation $y=\frac{0.83}{1+e^{2.27-0.36^{x}}}$ (Figure 1B). Calculations revealed that the nodes of the curve were at *x* = 0.5 and *x* = 12.05 [47]; that is, the curve was in the fast growth phase when the ratio of irradiated to non-irradiated males was between 0.5:1 and 12.05:1, and it reached the slow growth phase when the ratio was higher than 12.05:1. The infertility rate at this node was 73.53% when the ratio of irradiated males to non-irradiated males was 12.05:1.

### 3.2. Selection of the Age of Irradiated Males at Release

One-way ANOVA showed that when the ratio of irradiated to non-irradiated males was 12:1, the ages of the irradiated males had no significant effect on the number of eggs laid by females (*F*_3,56_ = 2.291, *p* = 0.088, Figure 2) or on the infertility rate (*F*_3,56_ = 2.373, *p* = 0.080, Figure 2). 

### 3.3. Control Effects of Irradiated Males after Release in the Field Cages

On the third and fifth days after the release of *S. frugiperda* in the field cages, there was no significant difference in hatching rates between the different treatment plots (*F*_5,18_ = 0.796, *p* = 0.567; *F*_5,18_ = 0.854, *p* = 0.530, Table 1). However, on days 7, 9, 14, and 21 after release, there were significant differences in the hatching rates between different treatment plots (*F*_5,18_ = 4.906, *p* = 0.005; *F*_5,18_ = 6.358, *p* = 0.001; *F*_5,18_ = 7.350, *p* = 0.001; *F*_5,18_ = 7.350, *p* = 0.001, Table 1). There was no significant difference in the total number of eggs laid by females between the treatment plots (*F*_5,18_ = 0.553, *p* = 0.735, Table 1). On the seventh day after release, the hatching rates of the plots with 1:0 and 20:1 irradiated males to non-irradiated males were 7.54% and 14.24%, respectively, which were significantly lower than that of the blank control (47.37%), and there was no significant difference between the plot with the 20:1 ratio of irradiated males and non-irradiated males and the 1:0 treatment group (7.54%). On day nine after release, the hatching rates in the plots with 1:0, 12:1, and 20:1 irradiated males to non-irradiated males (17.82%, 29.89%, 25.89%) were significantly lower than that of the blank control (76.69%), and there was no significant difference when the release ratio was 12:1, 20:1, and 1:0. On days 14 and 21 after release, the hatching rates in the plots with irradiated males (18.98–40.18%) were significantly lower than that of the blank control (79.01%) when the ratios of irradiated males to non-irradiated males were 1:0, 12:1, 16:1, and 20:1 (Table 1).

On the third, fifth, and seventh, days after the release of *S. frugiperda*, there was no significant difference in the leaf damage index between the different treatment plots (*F*_5,18_ = 0.85, *p* = 0.532; *F*_5,18_ = 0.786, *p* = 0.573; *F*_5,18_ = 1.701, *p* = 0.185, Figure 3A–C). However, on the ninth, fourteenth, and twenty-first days after the release of *S. frugiperda*, there were significant differences in the leaf damage index in the different treatment plots (*F*_5,18_ = 3.956, *p* = 0.014; *F*_5,18_ = 5.382, *p* = 0.003; *F*_5,18_ = 9.328, *p* < 0.001, Figure 3D–F). On day nine after release, the leaf damage index for the 1:0:1, 16:1:1, 20:1:1, and insecticide-treated plots (0.65–0.97%) was significantly lower than that for the control plots (2.38%) (Figure 3D). On the fourteenth day after release, the leaf damage index for all treatment plots (4.24–9.13%) was significantly lower than that for the control (22.28%) (Figure 3E). On the twenty-first day after release, the leaf damage index for all treatment plots (8.27–18.35%) was also significantly lower than that for the control (37.92%) (Figure 3F).

On the 21st day after insect release, there were no significant differences in the corrected non-hatching rate of eggs, the corrected population decline rate, or the corrected leaf protection rate among the different treatments (*F*_4,15_ = 0.821, *p* = 0.533, Figure 4A; *F*_4,15_ = 0.991, *p* = 0.444, Figure 4B; *F*_4,15_ =1.562, *p* = 0.235, Figure 4C).

## 4. Discussion

In this study, male pupae of *S. frugiperda* were irradiated with 250 Gy of X-rays, and both irradiated and non-irradiated males were paired with non-irradiated females in different ratios to investigate the appropriate release ratio of the irradiated males. The logistic curve fitting results showed that a 12:1 ratio of irradiated to non-irradiated males achieved an ideal control effect, with a theoretical infertility rate of 74%. By using 1–4-day-old irradiated males to compete against 1-day-old non-irradiated males at a ratio of 12:1, it was shown that the ages of the irradiated males did not have a significant effect on their competitive ability. The results of the field experiments showed that when the ratio of irradiated to non-irradiated males was 12:1–20:1, the rate of larval hatching in the field was significantly reduced, the insect decline rate was 58–83%, the leaf protection rate was 48–69%, and the control effect was nearly as high as that achieved with insecticide (insect decline rate = 93%; leaf protection rate = 77%).

It is essential to calculate the number of irradiated insects required for release when using the sterile insect technique. To ensure pest control, the ratio of irradiated to non-irradiated males should be established before the widespread release of sterile insects [48,49]. Currently, there are several studies that have evaluated the release ratio of irradiated insects. Hight et al. showed that a release ratio of 5:1 between sterile and normal males of *Cactoblastis cactorum* (cactorum) provided good control effects [50]. Steiner (1969) noted that a 20:1 release ratio best controlled *Bactrocera dorsalis* (Hendel), *B. cucurbitae*, and *C. capitata* populations [51]. Likewise, a study by Bond et al. (2021) showed that a 10:1 release ratio had a significant inhibitory effect on egg hatching in *A. aegypti* and *A. albopictus* [40], and another study showed an ideal ratio of 5:1 of irradiated to non-irradiated *Halyomorpha halys* (Stal) males [52]. In this experiment, the logistic fitting results showed that the release ratio of irradiated and non-irradiated males of *S. frugiperda* should be 12:1 and the infertility rate reached 73.53% at this release ratio. The release ratio should also be checked in the field after the ratio has been initially determined, given that it is only a theoretical conclusion from laboratory experiments and that the ecological conditions in the field are more complex [53].

Field release of irradiated insects usually requires a higher release ratio to achieve better control when compared to the ratio determined by laboratory experiments. For example, when *Anastrepha fraterculus* (Wiedemann) was exposed to a 40 Gy dosage of X-rays, the control effect was ideal when the ratio of irradiated to non-irradiated insects reached 50:1 [44]. When the release ratio of sterile to normal male medflies was 100:1, the induced sterility rate exceeded 70% [54]. We used ratios of 12:1, 16:1, and 20:1 (irradiated: non-irradiated males) to evaluate *S. frugiperda* damage in the field, and we discovered no significant differences between the results of the 12:1, 16:1, and 20:1 release ratios and the ability to control the pests with insecticide. Theoretically, the more irradiated males released, the better the control of the pest population, the ratio of irradiated insects to wild insects is close to 1:0 in the case of excessive release of irradiated males, so the control effect of insects at this ratio can be regarded as the highest that can be achieved by using insect sterility technology. We found no significant difference in pest control when the release ratio was 12:1, 16:1, 20:1, or 1:0. This demonstrated feasibility by using a release ratio of 12:1–20:1 in SIT program of *S. frugiperda*. Infertility rates in the field were lower than those in laboratory experiments with the same release ratio, although significant control effects were observed after the release of the irradiated insects. This difference may be related to the greater sensitivity of females to the perception of male quality in the field environment; for example, *Anastrepha ludens* (Loew) had a better relative sterility index in indoor mating than in field mating [55]. As a result, it is important to observe the ability of irradiated insects to compete in their natural environment. Light, temperature, and humidity in the field may all play a role in how females assess the quality of a male [40,56,57].

In many species, the first 24 h after emergence are the most successful for females with respect to mating, but most females do not have an age preference for males when mating. However, some polyandrous species do have an age preference for males [58,59,60]. For example, *Drosophila pseudoobscura* (Frolova) females prefer to mate with older males, a behavior that results in higher fecundity [61]; the same mating preference for older males is also present in *Drosophila bipectinate* (Duda) [62]. In contrast, *Lutzomyia longipalpis* (Lutz & Neiva) females showed a high level of interest in middle-aged males [63]; *Colaphellus bowringi* (Baly) females also showed a preference for middle-aged males [64]. In addition, female insects can also prefer young males, such as *C. capitata* females [65,66]. In lepidopteran insects, many studies have shown a negative correlation between male age and female egg production [67]. For example, Meng et al. showed that female of *Dendrolimus punctatus* (Walker) preferred to mate with young or middle-aged (day of emergence or 2 d) males when choosing among 1–4-day-old males [68]. There is no current evidence that *S. frugiperda* females have a particular preference regarding the age of the male; however, one study showed that mating with a 4-day post-emergence male produced the highest egg production compared to mating with a 1-day post-emergence male [69]. In this study, 1–4-day-old irradiated males competed against 1-day-old non-irradiated males to explore the age of irradiated males used for field release. The results showed that, although the 4-day-old irradiated males had the highest egg production and sterility rates after competitive mating, there were no statistically significant differences compared to other age groups. This indicates that females had no age preference among irradiated males that were 1–4 days old after emergence. Therefore, in the implementation of sterile insect technology, the control effects of irradiated males with ages from 1 to 4 days old are similar, which may reduce the difficulty of raising insects in bulk.

In order to achieve better control effects, sterile insect techniques are usually combined with other types of pest control [70]. For example, Canada was successful in reducing *C. pomonella* damage by releasing sterile males over large areas following the use of insecticides to control *C. pomonella* numbers [71]. The southern United States and northern Mexico’s *P. gossypiella* eradication project have used transgenic Bt cotton, mating disruption, and the release of sterile moths [27]. In the 1990s, the United States, Brazil, and other countries used Bt corn to control *S. frugiperda* with excellent results [72,73,74,75,76]. However, due to the rapid development of resistance in *S. frugiperda*, the control effectiveness of Bt corn is decreasing annually and the use of chemical insecticides is still unavoidable [77,78]. In response to the invasion of *S. frugiperda*, the Chinese government has granted biosecurity certificates for genetically modified insect-resistant corn (http://www.moa.gov.cn/ztzl/zjyqwgz/, accessed on 20 December 2022). Li et al. (2022) examined the insecticidal effects of various types of genetically modified corn, and the results showed that the genetically modified corn had a significant mortality effect on major lepidopteran pests [79]. Liang et al. (2021) posited that transgenic Cry1Ab corn has great potential for the control of *S. frugiperda* [80]. The susceptibility of *S. frugiperda* to transgenic Bt corn is currently established for a wide geographical range of populations, and methods for monitoring resistance have been established in China [81]. This provides a basis for the combined use of sterile insect technology and Bt transgenic corn for insect control. For *S. frugiperda* populations in China, we suggest integrating insect-resistant Bt maize planting and environmentally friendly pesticide spraying with irradiated males of *S. frugiperda* releasing to suppress target pest populations in the annual breeding area of *S. frugiperda* in southern China. This approach not only reduces the cost of using sterile insect technologies but also inhibits the development of pest resistance. However, this study is only a theoretical exploration, which can provide a new idea for the management measures of *S. frugiperda*, but there is still a long distance between it and the practical application. 

Even though the optimal release ratio for irradiated males of *S. frugiperda* has been identified in this study, there are still many unresolved issues that must be addressed before widespread release. For example, despite the development of a low-cost artificial diet for *S. frugiperda* [82], more research is needed to determine whether this diet affects the adult’s ability to adapt to being in the wild and also how sensitive it is to X-rays. The sterile insect technology generally provides the benefits of safety, environmental friendliness, and a low recurrence rate compared to traditional control techniques. However, there is still much work to be done to ensure the successful development of such a project, including large-scale insect breeding and quality testing of the insects before release.

## 5. Conclusions

This study assessed the ability of irradiated *S. frugiperda* males to compete for mating. The ratio of irradiated to non-irradiated males reached 12:1 in both laboratory and field experiments, which led to the best control effect (74% sterility rate, 48% leaf protection rate, and 58% parasite decrease rate). Our results provide fundamental information for future prevention and control of *S. frugiperda* using sterile insect techniques.

## Figures and Tables

**Figure 1 insects-14-00137-f001:**
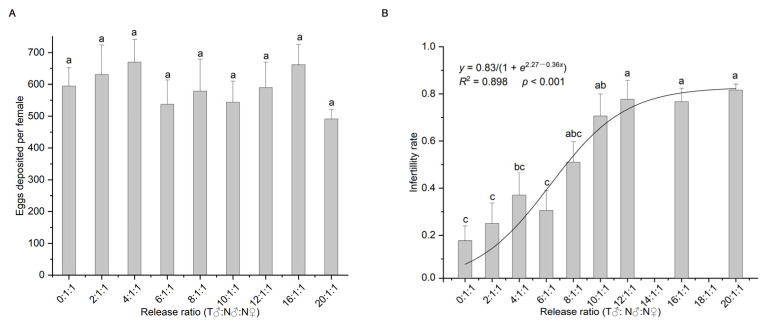
Average number of eggs laid per female (**A**), infertility rate (probability of non-hatching eggs) (mean ± standard error) and the fitting of the release ratio to the infertility rate (probability of non-hatching eggs) (**B**) in *Spodoptera frugiperda* at different release ratios; different lowercase letters indicate significant differences (one-way ANOVA, Tukey’s HSD; *p* < 0.05). T = treated with irradiation; N = not irradiated.

**Figure 2 insects-14-00137-f002:**
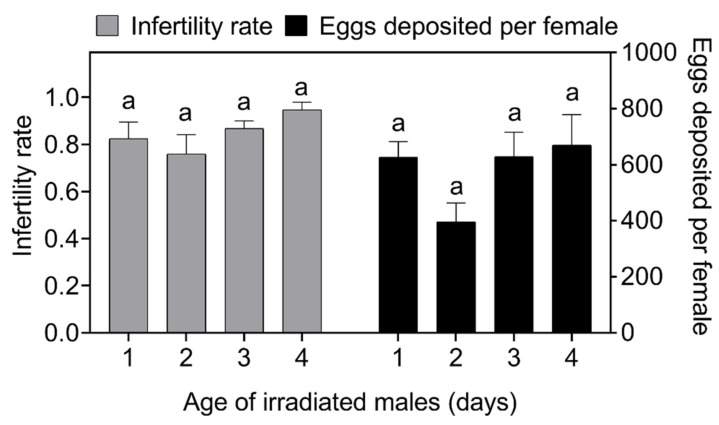
Effect of the ages of irradiated *Spodoptera frugiperda* males on egg production and the infertility rate (probability of non-hatching eggs). Data in the figure are the mean ± standard error; different lowercase letters indicate significant differences (one-way ANOVA, Tukey’s HSD; *p* < 0.05). In this experiment, the irradiated male: non-irradiated male: non-irradiated female ratio = 12:1:1, where both the non-irradiated male and the female were 1 day old.

**Figure 3 insects-14-00137-f003:**
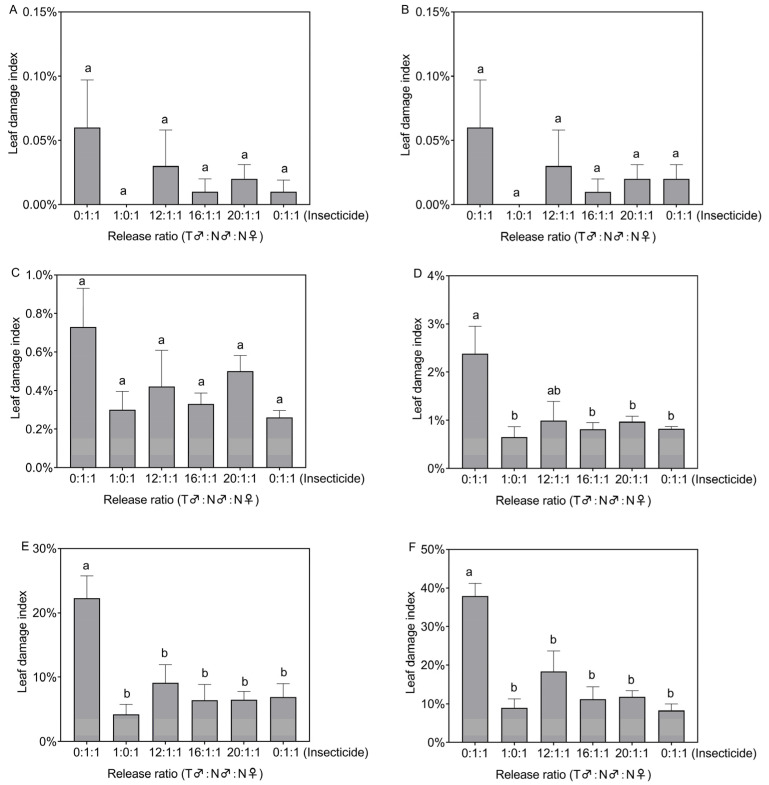
Leaf damage index in the treated plots on the 3rd (**A**), 5th (**B**), 7th (**C**), 9th (**D**), 14th (**E**), and 21st (**F**) days after *Spodoptera frugiperda* release. Data in the figure are the mean ± standard error; different lowercase letters indicate significant differences (one-way ANOVA, Tukey’s HSD; *p* < 0.05); T = treated with irradiation; N = not irradiated; insecticide was sprayed on the third day after insect release.

**Figure 4 insects-14-00137-f004:**
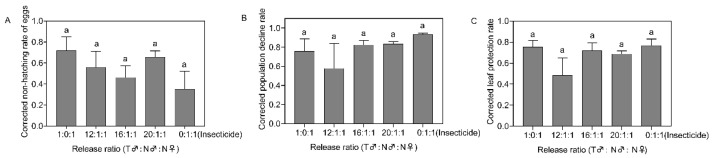
Corrected non-hatching rate of eggs (**A**), corrected population decline rate (**B**), and corrected leaf protection rate (**C**) after 21 days of field release of *Spodoptera frugiperda* in different proportions. Data in the figure are the mean ± standard error; different lowercase letters indicate significant differences (one-way ANOVA, Tukey’s HSD; *p* < 0.05); T = treated with irradiation; N = not irradiated; insecticide were sprayed on the third day after insect release.

**Table 1 insects-14-00137-t001:** Changes in the egg hatching rates in different treatment plots after the release of *Spodoptera frugiperda*.

Release Ratio	Hatching Rate of Eggs (%)						Total Number of Eggs
T♂:N♂:N♀ ^1^	3 Days after Release	5 Days after Release	7 Days after Release	9 Days after Release	14 Days after Release	21 Days after Release	
0:1:1 (blank control)	2.22 ± 1.34 a ^2^	0.89 ± 0.52 a	47.37 ± 8.38 a	76.69 ± 11.83 a	79.01 ± 10.24 a	79.01 ± 10.24 a	5447.00 ± 1306.96 a
1:0:1	0.00 ± 0.00 a	0.00 ± 0.00 a	7.54 ± 3.64 b	17.82 ± 6.57 b	18.98 ± 6.60 b	18.98 ± 6.60 b	3704.00 ± 760.05 a
12:1:1	0.43 ± 0.43 a	0.30 ± 0.30 a	20.61 ± 6.13 ab	29.89 ± 8.91 b	31.05 ± 9.15 b	31.05 ± 9.15 b	4225.50 ± 1620.53 a
16:1:1	0.16 ± 0.16 a	0.05 ± 0.05 a	19.59 ± 7.40 ab	39.13 ± 7.54 ab	40.18 ± 6.94 b	40.18 ± 6.94 b	3595.00 ± 927.59 a
20:1:1	2.19 ± 1.30 a	0.80 ± 0.46 a	14.24 ± 3.23 b	25.89 ± 6.79 b	27.57 ± 6.67 b	27.57 ± 6.67 b	4246.75 ± 696.65 a
0:1:1 (Insecticide) ^3^	7.01 ± 7.01 a	3.59 ± 3.50 a	33.43 ± 8.14 ab	43.70 ± 6.29 ab	46.04 ± 6.31 ab	46.04 ± 6.31 ab	3172.75 ± 713.26 a

^1^ T = treated with irradiation; N = not irradiated. ^2^ All data in the table are the mean ± standard error, and different lowercase letters in the same column represent significant differences in egg hatching rates between different plots (one-way ANOVA, Tukey’s HSD; *p* < 0.05). ^3^ Insecticide was sprayed on the third day after insect release.

## Data Availability

The data presented in this study are available on request from the corresponding author. The data are not publicly available due to privacy restrictions.

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
