# Peer review of "Mating Competitiveness of Male Spodoptera frugiperda (Smith) Irradiated by X-rays"

_insects, 2023, doi:10.3390/insects14020137_

Round 1
Reviewer 1 Report
The manuscript “Mating Competitiveness of Male Spodoptera frugiperda (J.E. Smith) Irradiated by X-rays” investigated a sterile insect technique for the permanent control of S. frugiperda and demonstrated the methods for their field application. This is an excellent study with clear and convincing data. The authors do an especially good job in placing their findings in the context of other research in the area. The experiment is very meaningful for the green control of fall armyworm. However, there were some questions listed as follows:
1. Only 250 Gy irradiation dose has been used in this experimental, however, different irradiation dose may have a different sterile insect result. Is an irradiation dose of 250 Gy the optimal dose? If not, is it possible to investigate the optimum radiation dose further? You know, the right radiation dose is very important to practical applications.
2. Eight-day-old male pupae were used in this study, why? Is there any basis or reference for the choice of time?
3. Does X-ray irradiation of pupae cause male deformities? Will the sexual competitiveness of irradiated male be affected by deformities?
4. The molecular mechanisms of how X-ray radiation cause infertility needs to be further investigated.
Author Response
The manuscript “Mating Competitiveness of Male Spodoptera frugiperda (J.E. Smith) Irradiated by X-rays” investigated a sterile insect technique for the permanent control of S. frugiperda and demonstrated the methods for their field application. This is an excellent study with clear and convincing data. The authors do an especially good job in placing their findings in the context of other research in the area. The experiment is very meaningful for the green control of fall armyworm. However, there were some questions listed as follows:
- Only 250 Gy irradiation dose has been used in this experimental, however, different irradiation dose may have a different sterile insect result. Is an irradiation dose of 250 Gy the optimal dose? If not, is it possible to investigate the optimum radiation dose further? You know, the right radiation dose is very important to practical applications.
Response: In our previous study, we found that 250 Gy of X-rays made the ideal sub-sterilization dose, and we have added the description of this study in the manuscript: “In our previous study [32], 8-day-old male pupae of S. frugiperda were exposed to X-rays at dosages ranging from 50 to 400 Gy, and 250 Gy dose had no significant effect on biological parameters such as emergence and life span of male pupa, but it could make the egg sterility rate reach 85%”. Lines 120-123.
- Eight-day-old male pupae were used in this study, why? Is there any basis or reference for the choice of time?
Response: The selection of 8-day-old pupae was based on references North (1975) and Osouli et al. (2021), and pupae close to emergence had a higher survival rate after irradiation.
Accordingly, we have added the above references in the manuscript “The used insects were male pupae that were 8 days old (1-2 days before emergence) [23,32,43] and exposed to radiation at ambient temperature.” Lines 153-155
23. North, D.T. Inherited sterility in Lepidoptera. Annu Rev Entomol 1975, 20, 167-182, doi:10.1146/annurev.en.20.010175.001123.
32. Jiang, S.; He, L.M.; He, W.; Zhao, H.Y.; Yang, X.M.; Yang, X.Q.; Wu, K.M. Effects of X‐ray irradiation on the fitness of the established invasive pest fall armyworm Spodoptera frugiperda. Pest Manag Sci 2022, 78, 2806-2815, doi:10.1002/ps.6903.
43. Osouli, S.; Ahmadi, M.; Kalantarian, N. Radiation biology and inherited sterility in Helicoverpa armigera Hübner (Lepidoptera: Nuctuidae). Int J Trop Insect Sc 2021, 41, 2421-2429, doi:10.1007/s42690-020-00418-y.
3.Does X-ray irradiation of pupae cause male deformities? Will the sexual competitiveness of irradiated male be affected by deformities?
Response: In our previous study, it was found that 250 Gy dose of X-ray had no significant effect on biological parameters such as emergence and life span of male pupa, the deformity rate of irradiated males was not significantly different from that of normal males, and we have added the description of this study in the manuscript: “In our previous study[32], 8-day-old male pupae of S. frugiperda were exposed to X-rays at dosages ranging from 50 to 400 Gy, and 250 Gy dose had no significant effect on biological parameters such as emergence and life span of male pupa, but it could make the egg sterility rate reach 85%.”. Lines 120-123.
- The molecular mechanisms of how X-ray radiation cause infertility needs to be further investigated.
Response: Thanks for your suggestion, this is good idea for future study.
Reviewer 2 Report
Insects-2145723
Title: Mating Competitiveness of Male Spodoptera frugiperda (J.E. Smith) Irradiated by X-rays
Summary
This manuscript uses a combination of laboratory and field experiments to prevent and control FAW by using males exposed to X-ray radiation to cause an increase in infertility rates. The research is very novel, but the experimental design, result presentation and practical application need to be improved, with emphasis on three aspects:
1. The aim of the authors' study was to investigate the competitiveness of irradiated males, but the design of the first experiment was flawed. At a ratio of 12:1, the irradiated males were themselves outnumbered by the unirradiated males, and thus had a higher chance of mating with females. The design itself had unconsciously increased the competitiveness of the irradiated males. The authors should have designed the experiment to ensure that the total number of males was consistent, Namely, a ratio for 0:21, 2:19, 4:17, 6:15, 8:13, 10:11, 12:9, 16:5, and 20:1 rather than 0:1, 2:1, 4:1, 6:1, 8:1, 10:1, 12:1, 16:1 and 20:1.
2. According to the data in Table 1, when the ratio of irradiated males to unirradiated males to females was 1:0:1, the egg hatching rate of FAW was the lowest, which did not exceed 20%. This suggests that infertility rates are highest when only irradiated males mate with females. But the authors must include an unirradiated male in their discussion, which I don't quite understand and hope the authors can clarify further.
3. In terms of practical application, authors also need to consider costs. First, to control one female (and her offspring), it is costly to raise more than 12 irradiated males. And the release of too many FAW adults can cause environmental stress. The cost of pesticides to kill one female is extremely low.
In addition, the release of irradiated males is only a short-term cure and does not produce heritable "harmful genes". In fact, insect infertility techniques often require specific conditions, and infertility is often exploited in order to pass on heritable genes. Oxitec, for example, has genetically engineered male Aedes aegypti mosquitoes to carry a self-limiting gene that is lethal to female Aedes aegypti. When these genetically modified males are released into the environment, they mate with wild females, and their female offspring die before reaching adulthood. The male offspring, half of whom still carry the self-limiting gene, continue to mate and kill the female offspring, and with each mating generation, the wild Aedes aegypti population will become smaller and smaller until it disappears. The X-ray radiation technique used in this study only destroys offspring and is not heritable, and there is no mechanism to support the study, let alone determine whether radiation causes the production of "self-limiting genes".
Specific comments
L137-140 The part about X-ray radiation needs to be rewritten. The authors do not clearly explain how or for how long to radiate the pupae. Whether the FAW is irradiated once the mature larvae pupate, or whether the male pupa of the eighth day of age is irradiated. This needs to be clear, and it is recommended that the authors elaborate on how to distinguish between female and male pupae, since few people can distinguish between male and female pupae in the pupal stage. This part can be added in the introduction section.
Also, why did the authors choose eight days of radiation and whether it was too long? Generally speaking, the pupa shell has a protective effect on FAW, while the radiation efficiency of pupa decreases. Direct radiation of adults for 2-3 days may produce the same effect. It is suggested that the authors should try to conduct a comparative test later, which may greatly improve the efficiency of practical application of X-ray.
L141 What is the total area of the test field.
L153 Here the authors inadvertently set two variables. The design seems to guarantee a one-to-one ratio of unirradiated males to females, but it increases the overall number of males. We know that an increase in the ratio of males to females affects the overall number of eggs laid, so a male-to-female ratio of 21:1 is definitely higher than a one-to-one ratio of eggs laid. And a ratio of 21 to 1 could lead to mating saturation, meaning that there is no guarantee that all 21 males can mate with one female. So a ratio of 0:21:1, 2:19:1, 4:17:1, etc., and 20:1:1, seems more reasonable.
L165 What is the age of the unirradiated male? Is it 1 day old? What is the significance of this study? The authors need to figure out whether they want to study the effect of age or the effect of radiation. Generally, both male and female FAW insects are produced in one batch of eggs, so the instars are similar. The authors hope to use infertility rates to extrapolate female selectivity towards males, which has its own flaws. irradiated males: non-irradiated males: non-irradiated females = 12:1:1, which predispositions females to choose the 12 irradiated males more frequently. The authors were able to determine female selectivity by recording the frequency and duration of mating with unirradiated and irradiated males (1:1), either by video recording or direct observation.
L175-183 What is the relationship between “(irradiated males: non-irradiated males: non-irradiated females = 0:1:1 (control), 1:0:1, 12:1:1, 16:1:1, 0:1:1 (spraying insecticide))” and “Ten non-irradiated females and ten non-irradiated males were eventually released”. Does it mean that the quantity released is amplified 10 times according to the above proportion? I said 24 cages were used, but only 20 cages were used for 4 repetitions.
L214 In fact, the authors could have removed the ratio of "1" for females and simply shown the ratio of irradiated males to non-irradiated males. It is also clear in the process of expression and easier to focus.
L231 The design of the experiment itself was problematic. In the material method, it was to study the selectivity of females to unirradiated males versus irradiated males of different days of age, which itself created two variables. The results of the study looked at the effect of age, suggesting that the authors readjust the test or remove it.
L242 According to the authors, hatching rates in field trials were lowest when the number of unirradiated males was zero, i.e., when the ratio of irradiated males to females was 1:1, and the results were most useful for control. So the authors probably shouldn't have included unirradiated males in their design. The study should also focus on the rate of leaf loss or corn loss, which is more important than the rate of infertility, which does not necessarily lead to high damage, since radiation is likely to cause birth defects.
L292 The aim of the authors' study here was to explore the competitiveness of irradiated males. But there was a flaw in the design. At a ratio of 12:1, the irradiated males themselves outnumbered the unirradiated males, and thus had a higher chance of mating with females. The authors should have designed the experiment to ensure that the total number of males was consistent, Namely, a ratio 0:21, 2:19, 4:17, 6:15, 8:13, 10:11, 12:9, 16:5, and 20:1, rather than 0:1, 2:1, 4:1, 6:1, 8:1, 10:1, 12:1, 16:1, and 20:1.
L302 As shown in Table 1, when the ratio of irradiated male to unirradiated male to female was 1:0:1, the egg hatching rate of FAW was the lowest, less than 20%. This suggests that infertility rates are best when only the irradiated males are involved. But the discussion must include unirradiated males, which I don't quite understand, and I hope the authors will clarify in the discussion.
L360 In terms of practical application, authors also need to consider costs. First, to control one female, more than a dozen irradiated males must be artificially created. This is very costly, and the release of too many FAW adults can cause environmental stress. In addition, the release of irradiated males does not have a good long-term control effect because the offspring are prevented directly.
Insect infertility techniques usually require specific conditions, and infertility is often heritable. Oxitec, for example, has genetically engineered male Aedes aegypti mosquitoes to carry a self-limiting gene that is lethal to female A. aegypti. When these genetically modified males are released into the environment, they mate with wild females, and their female offspring die before reaching adulthood. The male offspring, half of whom still carry the self-limiting gene, continue to mate and kill the female offspring, and with each mating generation, the wild A. aegypti population will become smaller and smaller until it disappears.
Author Response
Summary
This manuscript uses a combination of laboratory and field experiments to prevent and control FAW by using males exposed to X-ray radiation to cause an increase in infertility rates. The research is very novel, but the experimental design, result presentation and practical application need to be improved, with emphasis on three aspects:
- The aim of the authors' study was to investigate the competitiveness of irradiated males, but the design of the first experiment was flawed. At a ratio of 12:1, the irradiated males were themselves outnumbered by the unirradiated males, and thus had a higher chance of mating with females. The design itself had unconsciously increased the competitiveness of the irradiated males. The authors should have designed the experiment to ensure that the total number of males was consistent, Namely, a ratio for 0:21, 2:19, 4:17, 6:15, 8:13, 10:11, 12:9, 16:5, and 20:1 rather than 0:1, 2:1, 4:1, 6:1, 8:1, 10:1, 12:1, 16:1 and 20:1. 1.
Response: Revised. In the current study, we aimed to know how to field release the irradiated male moths in the context of actual field situation to control FAW by using sterile insect technology. The principle of the sterile insect technique is to control pest populations by releasing large numbers of irradiated males to make it difficult for wild females to find normal males. We considered that the ratio of female to male in the wild is fixed to 1:1, and releasing more irradiated males can increase the mating probability of irradiated males. Hence, we designed to release 2, 4, 6, 8, 10, 12, 16, 20 irradiated males with 1 normal male and 1 female and evaluated the sterility rates in order to find a suitable release density. Sure, it is important to design experiment with the consistent total number of males to investigate the competitiveness of irradiated males, and we will perform the experiment in future studies.
Moreover, there are similar experimental designs in other insect competitive mating experiments to find the critical value of release density, and we have added some references to the manuscript as follows:
“Mastrangelo, T.; Kovaleski, A.; Botteon, V.; Scopel, W.; Costa, M.D.L.Z. Optimization of the sterilizing doses and overflooding ratios for the South American fruit fly. Plos One 2018, 13, e201026, doi:10.1371/journal.pone.0201026.
Shelly, T.E.; McInnis, D.O.; Rendon, P. sterile insect technique and the Mediterranean fruit fly: assessing the utility of aromatherapy in large field enclosures. Entomol Exp Appl 2005, 116, 199-208, doi:10.1111/j.1570-7458.2005.00328.x.”
2. According to the data in Table 1, when the ratio of irradiated males to unirradiated males to females was 1:0:1, the egg-hatching rate of FAW was the lowest, which did not exceed 20%. This suggests that infertility rates are highest when only irradiated males mate with females. But the authors must include an unirradiated male in their discussion, which I don't quite understand and hope the authors can clarify further.
Response: Revised. S. frugiperda had the lowest hatching rate (18.98%) when the ratio of irradiated to non-irradiated males was 1:0, while the hatching rate of non-irradiated insects was 79.01%. But in practice, we can't kill all the wild males and leave only the females. In theory, the ratio of irradiated insects to wild insects is close to 1:0 in the case of excessive release of irradiated males, so the control effect of insects at this ratio can be regarded as the highest that can be achieved by using insect sterility technology. We describe it in more detail in the discussion section of the manuscript, as follows: “Theoretically, the more irradiated males released, the better the control of the pest population, the ratio of irradiated insects to wild insects is close to 1:0 in the case of excessive release of irradiated males, so the control effect of insects at this ratio can be regarded as the highest that can be achieved by using insect sterility technology. We found no significant difference in pest control when the release ratio was 12:1, 16:1, 20:1 or 1:0. This demonstrated feasibility by using release ratio of 12:1–20:1 in SIT program of S. frugiperda.” Lines 354-360
- In terms of practical application, authors also need to consider costs. First, to control one female (and her offspring), it is costly to raise more than 12 irradiated males. And the release of too many FAW adults can cause environmental stress. The cost of pesticides to kill one female is extremely low.
Response: Thanks for your valuable suggestion. Sure, it is costly to raise more than 12 irradiated males to control one female. So, we would suggest that the sterile insect technology should not be used alone on a large scale, Bt maize adoption and pesticides spraying (pesticides used at a decreased level as compared to normal) could be used to reduce the pest population in combination with SIT, especially before releasing sterile insects, so as to reduce the release number of irradiated males and the related cost. We clarify this point in the discussion of the manuscript:
“For S. frugiperda population in China, we suggest to integrate insect-resistant Bt maize planting and environment-friendly pesticide spraying with irradiated males of S. frugiperda releasing to suppress pest population in the annual breeding area of S. frugiperda in southern China. This approach not only reduces the cost of using sterile insect technologies, but also inhibits the development of pest resistance.” Moreover, we will try to reduce field cost by optimizing irradiation dose, increasing sterility rates or lowering the releasing number in the future studies. Lines 413-418
In addition, the release of irradiated males is only a short-term cure and does not produce heritable "harmful genes". In fact, insect infertility techniques often require specific conditions, and infertility is often exploited in order to pass on heritable genes. Oxitec, for example, has genetically engineered male Aedes aegypti mosquitoes to carry a self-limiting gene that is lethal to female Aedes aegypti. When these genetically modified males are released into the environment, they mate with wild females, and their female offspring die before reaching adulthood. The male offspring, half of whom still carry the self-limiting gene, continue to mate and kill the female offspring, and with each mating generation, the wild Aedes aegypti population will become smaller and smaller until it disappears. The X-ray radiation technique used in this study only destroys offspring and is not heritable, and there is no mechanism to support the study, let alone determine whether radiation causes the production of "self-limiting genes".
Response: Thank you for your suggestion. In our previous study (Jiang et al., 2022), the sterility of X-ray-irradiated FAW pupae could get to the F2 generation, and the Net reproductive rate of F1 generation decreased by 92.7% compared with the normal insect population. The Net reproductive rate of F2 generation decreased by 62.17%-79.63% compared with the normal insect population.
Releasing genetic engineered-insects carrying self-limiting genes is a very effective means of insect control, but considering the irreversibility of this method, it will take a long time to test before it can be used in practice. Meanwhile, due to the strong migratory ability of S. frugiperda, this approach may lead to the large-scale spread of self-limiting genes, leading to the extinction of the species. At the same time, we plan to carry out studies on the mechanism of X-ray radiation to insects in the future in order to find better prevention and control methods.
Specific comments
L137-140 The part about X-ray radiation needs to be rewritten. The authors do not clearly explain how or for how long to radiate the pupae. Whether the FAW is irradiated once the mature larvae pupate, or whether the male pupa of the eighth day of age is irradiated. This needs to be clear, and it is recommended that the authors elaborate on how to distinguish between female and male pupae, since few people can distinguish between male and female pupae in the pupal stage. This part can be added in the introduction section.
Response: We appreciated your suggestion and have rewritten the part about X-ray radiation. Specifically, we have added a description of insects irradiated by X-ray irradiators to the manuscript. In Section 2.1, we also describe in detail how to distinguish between male and female pupae.
“The insects were illuminated on a lead table in the center of the irradiator. The irradiation was stopped and samples were collected when the total cumulative radiation dosage reached 250 Gy (3212 s). The used insects were male pupae that were 8 days old (1-2 days before emergence) [23,32,43] and exposed to radiation at ambient temperature.” Lines 151-155
“The female pupa has a median longitudinal slit on the 8th abdominal sternite, while both sides are flat, without any protuberances; the male pupa has the 8th abdominal sternite unsplit, and the 9th sternite with a median longitudinal slit and a semicircular protuberance on either side.” Lines 136-140
Also, why did the authors choose eight days of radiation and whether it was too long? Generally speaking, the pupa shell has a protective effect on FAW, while the radiation efficiency of pupa decreases. Direct radiation of adults for 2-3 days may produce the same effect. It is suggested that the authors should try to conduct a comparative test later, which may greatly improve the efficiency of practical application of X-ray.
Response: Thank you very much for your valuable advice. It is relatively easier to irradiate pupae than to irradiate adults, but thanks again, we will try to irradiate adults in future experiments. The selection of 8-day-old pupae was based on some references to which we have added references in the manuscript “The used insects were male pupae that were 8 days old (1-2 days before emergence) [23,32,43] and exposed to radiation at ambient temperature.” Lines 151-153
L141 What is the total area of the test field.
Response: The total area of the experimental field was 0.618 hm2. Lines 160-161
L153 Here the authors inadvertently set two variables. The design seems to guarantee a one-to-one ratio of unirradiated males to females, but it increases the overall number of males. We know that an increase in the ratio of males to females affects the overall number of eggs laid, so a male-to-female ratio of 21:1 is definitely higher than a one-to-one ratio of eggs laid. And a ratio of 21 to 1 could lead to mating saturation, meaning that there is no guarantee that all 21 males can mate with one female. So a ratio of 0:21:1, 2:19:1, 4:17:1, etc., and 20:1:1, seems more reasonable.
Response: We considered that the ratio of female to male in the wild is fixed to 1:1, and releasing more irradiated males can increase the mating probability of irradiated males. Therefore, this experiment was designed with reference to the actual situation of field release. Similarly, there are similar experimental designs in other insect competitive mating experiments, and we have added some references to the manuscript as follows:
“Mastrangelo, T.; Kovaleski, A.; Botteon, V.; Scopel, W.; Costa, M.D.L.Z. Optimization of the sterilizing doses and overflooding ratios for the South American fruit fly. Plos One 2018, 13, e201026, doi:10.1371/journal.pone.0201026.
Shelly, T.E.; McInnis, D.O.; Rendon, P. sterile insect technique and the Mediterranean fruit fly: assessing the utility of aromatherapy in large field enclosures. Entomol Exp Appl 2005, 116, 199-208, doi:10.1111/j.1570-7458.2005.00328.x.”
In addition, sterile insect techniques aim to reduce the likelihood of successful reproduction of natural population members of the same species by releasing large numbers of sterile insects into the environment; therefore, the program does not require all released sterile males to mate with females. Sterile males are less competitive than or equal to normal males, so the release of sterile males tends to be larger. Thus, mating saturation in females may be inevitable.
L165 What is the age of the unirradiated male? Is it 1 day old? What is the significance of this study? The authors need to figure out whether they want to study the effect of age or the effect of radiation. Generally, both male and female FAW insects are produced in one batch of eggs, so the instars are similar. The authors hope to use infertility rates to extrapolate female selectivity towards males, which has its own flaws. irradiated males: non-irradiated males: non-irradiated females = 12:1:1, which predispositions females to choose the 12 irradiated males more frequently. The authors were able to determine female selectivity by recording the frequency and duration of mating with unirradiated and irradiated males (1:1), either by video recording or direct observation.
Response: Non-irradiated insects were all 1 day old (Line 186), although pairs of females mated with irradiated males had the lowest hatching rate. However, in the field, wild male and female insects occurs at the same time and are of similar age. This experiment was designed to simulate the field situation and explore the age of irradiated insects required for field release. We have changed the title of section 2.3.2 from " Effect of the age of the irradiated males on the sterility of the offspring " to" Selection of the age of irradiated males at release" for better expression. Line 183, 253
L175-183 What is the relationship between “(irradiated males: non-irradiated males: non-irradiated females = 0:1:1 (control), 1:0:1, 12:1:1, 16:1:1, 0:1:1 (spraying insecticide))” and “Ten non-irradiated females and ten non-irradiated males were eventually released”. Does it mean that the quantity released is amplified 10 times according to the above proportion? I said 24 cages were used, but only 20 cages were used for 4 repetitions.
Response: The field release was divided into the following treatments: irradiated males: non-irradiated males: non-irradiated females = 0:1:1 (control), 1:0:1, 12:1:1, 16:1:1, 20:1:1, 0:1:1 (spraying insecticide), and each treatment was repeated four times. “The number of insects released from each treatment was 0:1:1 (control) = 0:10:10, 1:0:1 = 10:0:10, 12:1:1 = 120:10:10, 16:1:1 = 160:10:10, 20:1:1 = 200:10:10, 0:1:1 (spraying insecticide) = 0:10:10.” We have supplemented this part of the description in the manuscript. Lines 201-203
L214 In fact, the authors could have removed the ratio of "1" for females and simply shown the ratio of irradiated males to non-irradiated males. It is also clear in the process of expression and easier to focus.
Response: Agreed. We removed the "1" ratio for females from the result. Lines 239-240
L231 The design of the experiment itself was problematic. In the material method, it was to study the selectivity of females to unirradiated males versus irradiated males of different days of age, which itself created two variables. The results of the study looked at the effect of age, suggesting that the authors readjust the test or remove it.
Response: Agreed. This experiment was designed to simulate the field situation and explore the age of irradiated insects required for field release. Therefore, we have changed the title of section 2.3.2 from " Effect of the age of the irradiated males on the sterility of the offspring " to" Selection of the age of irradiated males at release". Lines 183, 253
L242 According to the authors, hatching rates in field trials were lowest when the number of unirradiated males was zero, i.e., when the ratio of irradiated males to females was 1:1, and the results were most useful for control. So the authors probably shouldn't have included unirradiated males in their design. The study should also focus on the rate of leaf loss or corn loss, which is more important than the rate of infertility, which does not necessarily lead to high damage, since radiation is likely to cause birth defects.
Response: Thanks for your suggestion, we calculated the killing rate of maize leaves in Figures 3 and 4, as well as the corrected leaf protection rate under different treatments. The effect of the release ratio on the control of insects in the laboratory experiment was presented by the infertility rate of eggs, so we also calculated the infertility rate of insects under different treatments in the field to compare with the effect of the laboratory experiment.
L360 In terms of practical application, authors also need to consider costs. First, to control one female, more than a dozen irradiated males must be artificially created. This is very costly, and the release of too many FAW adults can cause environmental stress. In addition, the release of irradiated males does not have a good long-term control effect because the offspring are prevented directly.
Response: Revised. The sterile insect technique has been used to control many invasive insect pests. Based on previous experience, we believe that planting Bt maize and releasing irradiated males may be effective for long-term pest control. When the sterile insect technology is implemented on a large scale, we will use insecticides and plant Bt corn to reduce the pest population before releasing sterile insects, so as to reduce the cost of release. We clarify this point in the discussion of the manuscript. In addition, according to the experience of previous studies, the release of sterile males is usually long-term and cyclical. The first release requires a large number of sterile insects to ensure that the pest population drops to the desired value, and then only a small number of sterile males are released periodically to achieve the purpose of controlling the pest population.
“For S. frugiperda population in China, we suggest to integrate insect-resistant Bt maize planting and environment-friendly pesticide spraying with irradiated males of S. frugiperda releasing to suppress pest population in the annual breeding area of S. frugiperda in southern China. This approach not only reduces the cost of using sterile insect technologies, but also inhibits the development of pest resistance.” Lines 409-414
Insect infertility techniques usually require specific conditions, and infertility is often heritable. Oxitec, for example, has genetically engineered male Aedes aegypti mosquitoes to carry a self-limiting gene that is lethal to female A. aegypti. When these genetically modified males are released into the environment, they mate with wild females, and their female offspring die before reaching adulthood. The male offspring, half of whom still carry the self-limiting gene, continue to mate and kill the female offspring, and with each mating generation, the wild A. aegypti population will become smaller and smaller until it disappears.
Response: Thank you for your suggestion. We believe that using genetic engineering to make insects carry self-limiting genes is a very effective means of insect control, but considering the irreversibility of this method, it will take a long time to test before it can be used in practice. Meanwhile, due to the strong migratory ability of S. frugiperda, this approach may lead to the large-scale spread of self-limiting genes, leading to the extinction of the species. At the same time, we plan to carry out studies on the mechanism of X-ray radiation to insects in the future in order to find better prevention and control methods.
Reviewer 3 Report
I reviewed the manuscript insects-2145723 (Mating competitiveness of male Spodoptera frugiperda irradiated by X-ray). I found the approach very interesting and definitely the experiment is providing very important data for the managing of this alien pest.
For my point of view, the authors should improve the introduction in some aspects. Definitely, the biology of the target pest is not well describe: and the correct application of a strategy such as SIT is requiring a very well background on the biology and mating/oviposition behavior of the pest.
Regarding the experimental design, the most critical aspects are in the test 2.31. It is not clear to me the size and structure of the "rearing bucket" (see my comments in the PDF): this is important because you are confining in this structure in choice conditions different ratios of irradiated and fertile males with fertile females.
I am wondering if the lack of space can make some side-effects.
Also is not clear the real numbers of adults per "rearing bucket".
How long last the experiment?
I guess there was a mortality among the adults: how the authors manage this? Did they mark the fertile (or irradiated) males?
For the confined field tests, I am wondering if the authors recorded differences in the fecundity (number of deposited eggs) when the number of males was increasing..
Finally, please check the bibliography (occasional format mistakes).

Author Response
I reviewed the manuscript insects-2145723 (Mating competitiveness of male Spodoptera frugiperda irradiated by X-ray). I found the approach very interesting and definitely the experiment is providing very important data for the managing of this alien pest.
For my point of view, the authors should improve the introduction in some aspects. Definitely, the biology of the target pest is not well describe: and the correct application of a strategy such as SIT is requiring a very well background on the biology and mating/oviposition behavior of the pest.
Response: Accepted. we added the following content to the manuscript: “The fall armyworm, Spodoptera frugiperda (Smith) (Lepidoptera: Noctuidae), is a tropical and subtropical pest native to the America that can damage corn, wheat, and other major crops [1,2], is well known by its polyphagia, high fecundity, and strong migration ability. During the reproductive period, female S. frugiperda can mate with different males and lay eggs [1]. A female S. frugiperda can lay about 1000 eggs in 3 to 7 days, and at the proper temperature, these eggs hatch in 2 to 4 days. The damage to the host plant is made worse as the larvae reach their sixth instar because their greedy appetite is completely displayed [3].” Lines 40-47
Regarding the experimental design, the most critical aspects are in the test 2.31. It is not clear to me the size and structure of the "rearing bucket" (see my comments in the PDF): this is important because you are confining in this structure in choice conditions different ratios of irradiated and fertile males with fertile females.
Response: Revised. The feeding bucket size is 15 L (diameter of 25 cm and height of 30 cm), we have added this in the materials and methods (Lines 142,173,188-189). In this study, all containers used for laboratory experiments were of the same size, and no significant difference was found in the number of eggs laid by females at different release ratios, which may explain that the density of adults in this experiment had no effect on the number of eggs laid by females. We added Fig. 1A to explain this result.
I am wondering if the lack of space can make some side-effects.
Response: revised. Larval density of S. frugiperda has been reported to affect female fecundity, but the effect of adult density on reproduction has not been investigated. In this study, all containers used for laboratory experiments were of the same size, and no significant difference was found in the number of eggs laid by females at different release ratios, which may explain that the density of adults in this experiment had no effect on the number of eggs laid by females. We added Fig. 1A to explain this result.
Also is not clear the real numbers of adults per "rearing bucket".
Response: Revised. There was one non-irradiated female and one non-irradiated male insect in each treatment, the irradiated males were released proportionally based on this. We added this to the manuscript in lines 175-177.
How long last the experiment?
Response: Revised. The laboratory experiment was conducted until all tested insects died, and the field experiment started on 7 September 2022 and ended on 30 September 2022. We added that to the manuscript. Lines 166-167
I guess there was a mortality among the adults: how the authors manage this? Did they mark the fertile (or irradiated) males?
Response: According to our previous study (Jiang et al., 2022), there was no significant difference in the lifespan between irradiated and normal males after 250 Gy of X-ray irradiation. This was confirmed again in the present study, during the oviposition period of the female, most of the males survived, and only a few individuals died.
Jiang, S.; He, L.M.; He, W.; Zhao, H.Y.; Yang, X.M.; Yang, X.Q.; Wu, K.M. Effects of X‐ray irradiation on the fitness of the established invasive pest fall armyworm Spodoptera frugiperda. Pest Manag Sci 2022, 78, 2806-2815, doi:10.1002/ps.6903.
For the confined field tests, I am wondering if the authors recorded differences in the fecundity (number of deposited eggs) when the number of males was increasing.
Response: We recorded the number of eggs laid by females, and this data has been supplemented in Table 1.
Finally, please check the bibliography (occasional format mistakes).
Response: Agreed. We rechecked the format of the references and corrected the errors.
Line 40: Author's name.
Response: Agreed. We have changed " Spodoptera frugiperda" to" Spodoptera frugiperda (Smith)". Line 40
Line 93: Please re-write the full genus name at the beginning of the sentence.
Response: Agreed. We have changed " S. frugiperda" to" Spodoptera frugiperda". Line 99
Lines 96-100: This part is very important. However it is not clear if the genetically modified maize and Bt maize are synonymous: please improve this part.
Response: Agreed. We have changed " genetically modified maize " to" Bt maize " Line 102
Lines 103-104: This is not always true (maybe it is correct for Lepidoptera): please improve the sentence.
Response: Agreed. We have changed "Research has shown that irradiated males are less competitive than normal males." to "Studies have shown that males exposed to high doses of irradiation in some species are less competitive than normal males, however, this competitive difference can be compensated for by increasing the release ratio." Lines 109-111
Line 105: 1971
Response: Agreed. We have changed " 1970" to" 1971". Line 112
Line 108: ...to previous studies on mosquitoes, infertile.
Response: Agreed. We have changed " According to previous studies…" to" According to previous studies on mosquitoes…". Lines 114-115
Line 156: Authors should provide additional information on this rearing system: size, material, climatic conditions, etc
Response: Accepted. We added this sentence in Section 2.3.1: “The feeding methods and the environment in which the insects were kept were as described in Section 2.1.” Lines 177-178
Round 2
Reviewer 2 Report
insects-2145723 R1
Title: Mating Competitiveness of Male Spodoptera frugiperda (J.E. Smith) Irradiated by X-rays
c
The manuscript has been greatly improved as suggested. But I still think the idea of using the mating of irradiated males to prevent FAW seems far-fetched. Radiation-induced mutations are unmanageable and hard to pass on to offspring to have widespread effects. It is suggested that the authors add some expressions to the discussion to minimize the practical significance of the experiment. The manuscript should indicate that this study is only a theoretical exploration, which can provide a new idea for the management measures of FAW, but there is still a long distance between it and the practical application. The manuscript may not be published until relevant expressions are added to the discussion.
Author Response
The manuscript has been greatly improved as suggested. But I still think the idea of using the mating of irradiated males to prevent FAW seems far-fetched. Radiation-induced mutations are unmanageable and hard to pass on to offspring to have widespread effects. It is suggested that the authors add some expressions to the discussion to minimize the practical significance of the experiment. The manuscript should indicate that this study is only a theoretical exploration, which can provide a new idea for the management measures of FAW, but there is still a long distance between it and the practical application. The manuscript may not be published until relevant expressions are added to the discussion.
Response: Thanks for your valuable suggestions. We added the following description to the manuscript: “However, this study is only a theoretical exploration, which can provide a new idea for the management measures of S. frugiperda, but there is still a long distance between it and the practical application.” Lines 417-420
Reviewer 3 Report
Please review the changes I suggested in the attached file. I am also suggesting to double check the English language and style.

Author Response
Please review the changes I suggested in the attached file. I am also suggesting to double check the English language and style.
Response: Agree with your suggestion. The English language and style have been checked.
Line 44 delete “and lay eggs”; A female of S. frugiperda...
Response: Accepted. We removed "and lay eggs" and replaced " A female S. frugiperda " with " A female of S. frugiperda ". Line 44
Line 45 Please be more precise: provide a trange of temperatures and bibliographic info.
Response: Agree with your suggestion. We have changed the sentence from " A female S. frugiperda can lay about 1000 eggs in 3 to 7 days, and at the proper temperature, these eggs hatch in 2 to 4 days." to " A female of S. frugiperda can lay about 1000 eggs in 4 to 9 days, and at the proper temperature (20°C-30°C), these eggs hatch in 2 to 4 days." Line 44-45
Line 135 Lowercase
Response: Accepted. Line 135
Line 178 change in "hatching rate"
Response: Accepted. We have changed " hatched" to " hatching rate ". Line 179
Line 191 Change in "Confined-field release test..."
Response: Accepted. We have changed "Field release test " to " Confined-field release test ". Line 192
Line 193: Please describe the size and the material of the cage.
Response: Agree with your suggestion, we have added the description of cage material and density in the manuscript. “Male pupae of S. frugiperda were irradiated with 250 Gy of X-rays and released proportionally to non-irradiated insects in a field cage making with 40 mesh white polyethylene after the irradiated males emerged.” Line 193-195
Line 262 and line 263 Field cage
Response: Accepted. We have changed " field" to " field cage ". Lines 263-264
Line 377 please write "female of Dendrolimus..." or "..showed that Dendrolimus punctatus (Walker) females...
Response: Accepted. We have changed " female Dendrolimus punctatus (Walker)" to " female of Dendrolimus punctatus (Walker) ". Lines 380
Line 409 If it is referred to just one population authors should write "For the S. frugiperda population in China"; viceversa, if is referred to more than one population, they should write "populations".
Response: Accepted. We have changed " For S. frugiperda population in China" to" For S. frugiperda populations in China ". Lines 412-413.
Line 412 As above: or "to suppress the target pest population" or "to suppress target pest populations..."
Response: Accepted. We have changed " to suppress pest population " to" to suppress target pest populations ". Line 415.
Line 422 delete “any”
Response: Accepted. Line 429
Line 523 and line 541 italics
Response: Accepted. Lines 529, 547